# Preparation of Naringenin Nanosuspension and Its Antitussive and Expectorant Effects

**DOI:** 10.3390/molecules27030741

**Published:** 2022-01-24

**Authors:** Zhengqi Dong, Rui Wang, Mingyue Wang, Zheng Meng, Xiaotong Wang, Meihua Han, Yifei Guo, Xiangtao Wang

**Affiliations:** 1Institute of Medicinal Plant Development, Chinese Academy of Medical Sciences & Peking Union Medical College, No. 151, Malianwa North Road, Haidian District, Beijing 100193, China; zqdong@implad.ac.cn (Z.D.); wmy313714384@163.com (M.W.); mz960428@163.com (Z.M.); ttwang0521@163.com (X.W.); hanmeihua727@163.com (M.H.); 2College of Pharmacy, Heilongjiang University of Chinese Medicine, Harbin 150040, China; wrdx@sina.com; 3College of Pharmacy, Harbin University of Commerce, No. 138, Tongda Street, Daoli District, Harbin 150076, China

**Keywords:** naringenin nanosuspension, media-milling method, bioavailability, antitussive effect, expectorant effect

## Abstract

Naringenin (NRG) is a natural flavonoid compound abundantly present in citrus fruits and has the potential to treat respiratory disorders. However, the clinical therapeutic effect of NRG is limited by its low bioavailability due to poor solubility. To enhance the solubility, naringenin nanosuspensions (NRG-NSps) were prepared by applying tocopherol polyethylene glycol succinate (TPGS) as the nanocarrier via the media-milling method. The particle size, morphology, and drug-loading content of NRG-NSps were examined, and the stability was evaluated by detecting particle size changes in different physiological media. NRG-NSps exhibited a flaky appearance with a mean diameter of 216.9 nm, and the drug-loading content was 66.7%. NRG-NSps exhibited good storage stability and media stability. NRG-NSps presented a sustainable release profile, and the cumulative drug-release rate approached approximately 95% within 7 d. NRG-NSps improved the antitussive effect significantly compared with the original NRG, the cough frequency was decreased from 22 to 15 times, and the cough incubation period was prolonged from 85.3 to 121.6 s. Besides, NRG-NSps also enhanced expectorant effects significantly, and phenol red secretion was increased from 1.02 to 1.45 μg/mL. These results indicate that NRG-NSps could enhance the bioavailability of NRG significantly and possess a potential clinical application.

## 1. Introduction

Naringenin (NRG), an aglycone of naringin, belongs to the dihydroflavones and mainly exists in natural plants such as citrus, aurantium, peach leaf, and grapefruit [1]. Studies have found that naringin has antibacterial [2], antioxidant and antiviral [3], and anticancer properties [4,5] that can also inhibit the expression of pro-inflammatory cytokines [6], which have potential clinical applications for therapy for pulmonary diseases such as *Staphylococcus aureus* pneumonia, pulmonary fibrosis, and asthma [7]. However, the solubility of naringin in water is very poor and almost insoluble, and the oral bioavailability of naringin is only 5.81%, which seriously affects its therapeutic effect and further restricts its clinical application [8].

In recent years, the development of nanotechnology has provided new ideas for improving the solubility and bioavailability of hydrophobic drugs [9,10]. Nanosuspensions, nanomicelles, nanoparticles, and nanocrystals are prepared with nanomaterials as nanocarriers to entrap hydrophobic drugs, and these nanodelivery systems could enhance the solubility of hydrophobic drugs in aqueous solution, further improving their bioavailability [11,12]. Nanosuspensions (NSps) present a smaller particle size, higher drug-loading content, simple preparation process, better stability after curing, and biological adhesion, which can significantly improve the bioavailability and efficacy of hydrophobic drugs [13,14,15]. NSps can be achieved via the rapid formation of crystal nuclei under controlled conditions, in which the growth of crystallization is inhibited by the stabilizer [16,17].

At present, there are three main methods for preparing nanosuspensions: the top-down method, bottom-up method, and a combination of the two [18]. The antisolvent precipitation method is attributed to a kind of bottom-up method: The hydrophobic drug is first dissolved in organic solvents and then rapidly mixed with distilled water. This process typically results in rapid particle growth and broad particle size distribution. Top-down technologies include the media-milling method and high-pressure homogenization. In the media-milling method, drug powder is added to a solution-containing surfactant. The drug particles and milling medium collide violently with the inner wall of the milling chamber to obtain nanoscale drug particles [19]. The advantages of the media-milling method are as follows: Drugs can be easily prepared as nanosuspensions; there is little difference between batches, it is easy to expand production, and it is suitable for industrial production; the preparation device is simple and easy to obtain; and the preparation only requires tens of milligrams of active pharmaceutical ingredients, which can effectively reduce the cost of scientific research. Dihydroartemisinin nanosuspension can be prepared using the milling method, and a previous study investigated its physical stability and antimalarial activity in vitro [20]. Furthermore, naringenin nanosuspension can be prepared using the milling method, which has made it easier for the drug to be absorbed in vivo, thereby improving its bioavailability [21].

Currently, a certain amount of amphiphilic compound is exploited and utilized as a stabilizer to prepare nanosuspension, which could reduce sedimentation, aggregation, crystal growth, crystal transformation, and other phenomena [22]. It is clear that different drugs present different properties, and different stabilizers should be used to ensure the successful preparation of nanosuspensions. Therefore, the selection of an appropriate stabilizer is of great importance [23]. Chitosan, polyvinylpyrrolidone (PVP), and other commonly used stabilizers have been used in the preparation of naringenin nanosuspensions (NRG-NSps) [24,25,26,27]. However, these NRG-NSps show certain drawbacks, such as low stability and poor bioavailability. To overcome these drawbacks, NRG-NSps should be prepared by choosing a stabilizer with good biological properties that can adapt to a variety of administration modes. Tocopherol polyethylene glycol vitamin E succinate (TPGS) is synthesized by coupling between the carboxyl group of vitamin E succinate and the hydroxyl group of polyethylene glycol 1000 (PEG1000), which is an amphiphilic medical excipient recognized by the Food and Drug Administration (FDA) [28], and can be used in pharmaceutical preparations as a solvent enhancer, emulsifier, and stabilizer [29]. In recent years, owing to the good solubility, permeability, and stability of TPGS, it has been utilized as nanocarriers to construct various nanodelivery systems, such as liposomes, micelles, nanoparticles, and nanocrystals [30,31].

In this study, using NRG as the model drug and TPGS as the stabilizer, NRG-NSps were prepared using the media-milling method, and the optimal formulation and process were determined through univariate analysis. To avoid deterioration after storage for a long time, NRG-NSps were made into lyophilized powder to ensure their storage stability. The particle size, morphology, stability, and drug-release characteristics of NRG-NSps were investigated. In addition, the expectorant effect of NRG-NSps was estimated with normal mice via the phenol red secretion method, and the antitussive effect in vivo was researched with ammonia-induced cough model mice.

## 2. Results and Discussion

### 2.1. Physicochemical Characteristics of NRG-NSps

Using the “top-down” principle, the media-milling method was applied to prepare NRG-NSps. The magnetic stirrer served as the power device, the vial was the milling chamber, and the zirconia ball was used as the milling medium to prepare NRG-NSps. The feed–weight ratio of NRG vs. TPGS was designed as 4/1. After stirring for 2 h, NRG-NSps was prepared successfully as a white milky liquid, and the drug-loading content was 66.7%. The particle size of NRG-NSps was 216.9 nm. The particle size distribution curve is shown in Figure 1a. Transmission electron microscopy (TEM) images revealed that NRG-NSps presented a flaky morphology (Figure 1b), which is different than the appearance of nanospherical particles reported in previous papers [32,33].

### 2.2. Stability of NRG-NSps

To estimate the media stability, NRG-NSps were incubated with different physiological media, including PBS, normal saline, glucose solution, artificial gastric juice, and artificial intestinal fluid. The particle size change curves are shown in Figure 2. After incubating for 24 h, the state of the nanosuspension solution showed no obvious change, no aggregation or precipitation was observed, and the particle size was maintained in all media. These results indicated that NRG-NSps presented excellent media stability in all test media, which was suitable for oral administration.

To verify the storage stability, NRG-NSps were stored at 4 °C for one month, and the appearance of the nanosuspension solution showed no significant change. The particle sizes are summarized in Table 1. After detection by dynamic light scattering (DLS), it was proven that the particle size and surface charge of NRG-NSps was maintained excellently, which was approximately 220 nm with a polydispersity index (PDI) of approximately 0.27 and −1.0 mV, relatively. There were no significant deviations in the dimensions or surface charge of the nanoparticles during the storage period, indicating good storage stability.

Due to the Ostwald ripening phenomenon, nanoparticles in solution aggregate to form larger particles, which could affect the stability [34]. For long-term storage, nanosuspension solution should be lyophilized to powder, and then reconstituted without significant change. In this study, we selected several freeze-dried protective agents at a concentration of 0.5%. When P188 was selected as the lyophilized protective agent, NRG-NSps could be lyophilized and reconstructed in water successfully, with a particle size of NRG-NSps of 223.7 ± 3.4 nm (PDI~0.18 ± 0.02). The particle size presented no significant difference from the initial value, and it was proven that NRG-NSps exhibited good lyophilized reconstitution stability.

### 2.3. Drug-Release Behaviour

The cumulative release rate of the NRG powder suspension was only 30.6% within 168 h, and exceeding 92%, NRG was released from NRG DMSO solution within 8 h, which were used as the control groups. NRG-NSps were detected under identical conditions. The nanosuspension prepared with TPGS as a stabilizer showed a moderate release rate. The cumulative release rate in vitro reached approximately 25% within the initial 8 h, then a slow sustained release was shown, with the cumulative release rate reaching approximately 95% in the following 160 h (Figure 3). Compared with NRG powder, NRG-NSps showed a higher release rate. This phenomenon was mainly attributed to the enhanced solubility of NRG in aqueous solution and larger surface area in terms of the small particle size of NRG-NSps. Compared with NRG DMSO solution, NRG-NSps presented a slower release rate and could be sustainably released for 7 d, which could be attributed to the structure of NRG-NSps. The release profile of NRG was affected by the steric hindrance of nanoparticles [35]. Based on these results, NRG-NSps could enhance the aqueous solubility of NRG significantly and exhibit a sustainable release profile, which could influence the bioavailability of NRG.

### 2.4. Antitussive Assay

Coughing is caused by secretions or foreign substances that stimulate the mucous membrane of the respiratory tract and produce coughing action through neural reflexes, removing secretions from the airways [36]. Ammonia stimulates the mucosa of the respiratory tract and causes coughing; therefore, the antitussive effects of NRG-NSps were estimated with a mouse cough model induced by aqueous ammonia [37]. The cough frequency and cough incubation time of each group were used as evaluation indexes to investigate the antitussive effect of NRG-NSps. The mice were divided into a saline group (blank model control), dextromethorphan hydrobromide group (positive control), NRG group, and NRG-NSps group (high, medium, and low). The concentration of positive drug was 15 mg/kg and of NRG was 30 mg/kg, and the dosage of the NRG-NSps group was 10, 30, and 50 mg/kg (NRG equivalent concentration).

In 5 min, the blank model group coughed 33 ± 5 times, the positive control group coughed 14 ± 4 times, and NRG group coughed 22 ± 4 times. The cough frequency of NRG-NSps was 25 ± 4, 15 ± 3, and 12 ± 3 times at 10, 30, and 50 mg/kg, respectively (Figure 4a). The positive control group showed excellent antitussive effects, and compared with the blank model group, the cough frequency decreased 2.3-fold (33 vs. 14, *p* < 0.001) and the inhibition cough rate was 57%, revealing that the cough model in mice was constructed successfully. All the NRG-NSps showed moderate to good antitussive effects. Compared with the blank model group, the cough frequency of NRG-NSps decreased significantly by 1.3-fold (33 vs. 25, *p* < 0.05), 2.2-fold (33 vs. 15, *p* < 0.001), and 2.7-fold (33 vs. 12, *p* < 0.001), and the cough inhibition rate was 24%, 54%, and 64% for three NRG-NSps, respectively. Besides, the NRG-NSps group presented antitussive effects in mice in a certain dose-dependent manner, which was promoted sharply initially and then slowly with increasing concentration of NRG. The antitussive effect was significantly enhanced with the increase in dosage of NRG-NSps from 10 to 30 mg/kg, and the cough frequency was decreased approximately 1.6-fold (25 vs. 16 times, *p* < 0.05). Further increasing the concentration to 50 mg/kg, the cough frequency was decreased approximately 2.1-fold (25 vs. 12, *p* < 0.01), but the cough frequency only decreased slightly (15 vs. 12, *p* > 0.05) and no significant difference was presented when the concentration of NRG was enhanced from 30 to 50 mg/kg. Considering the antitussive effects, economy, and side effects, the concentration of NRG could be selected as 30 mg/kg. Under the same dosage (30 mg/kg), compared with free NRG, the cough frequency of the NRG-NSps group was decreased 1.5-fold and the inhibition cough rate was enhanced from 33% to 54% (22 vs. 15, *p* < 0.05), and the antitussive effect was obviously higher than that of free NRG. Compared with positive drugs, NRG-NSps (30 mg/kg) showed a similar antitussive effect, and there was no significant difference in cough frequency (14 vs. 15, *p* > 0.05). Based on these results, it seems that the NRG-NSps group could serve as an effective therapy for acute cough.

Subsequently, the antitussive effect of NRG-NSps was further evaluated by investigating the cough incubation time. In 5 min, the cough incubation time of the saline group was 61.6 ± 7.6 s, of the positive drug group was 135.3 ± 12.6 s, and of the NRG group was 85.3 ± 8.2 s. The incubation period of NRG-NSps was 80.8 ± 7.7, 121.6 ± 11.3, and 129.3 ± 6.7 s at 10, 30, and 50 mg/kg, respectively (Figure 4b). NRG-NSps could promote the cough incubation period significantly compared with the blank model group (*p* < 0.05). The cough incubation period also showed dose dependence, and was prolonged with an increase in dosage. At the same dose (30 mg/kg), NRG-NSps significantly increased the cough incubation period and delayed the onset of cough by 36.3 s compared with the original NRG.

Through the comparison of cough frequency and cough incubation period, NRG-NSps showed an enhanced cough-relieving effect, significantly reduced the cough frequency, and prolonged the cough incubation period.

### 2.5. Expectorant Assay

Expectorant activity was evaluated using the phenol red secretion mouse model according to the regression line of the spectrophotometric response [38]. The content of phenol red was 0.65 ± 0.13 μg/mL in the saline group, 1.17 ± 0.15 μg/mL in the positive drug group, and 1.02 ± 0.14 μg/mL in the free NRG group, and the content of phenol red was 1.08 ± 0.18, 1.45 ± 0.17, and 1.76 ± 0.12 μg/mL at 10, 30, and 50 mg/kg, respectively (Figure 5). NRG-NSps had expectorant effects on the phenol red secretion mouse model in a dose-dependent manner, and the expectorant effect was significantly enhanced with the increase in dosage of NRG-NSps (*p* < 0.01). At the same dose (30 mg/kg), compared with NRG, the excretion of phenol red in the trachea of NRG-NSps increased by 42% (1.45 vs. 1.02 μg/mL), and the expectorant effect was obviously better than free NRG (*p* < 0.05). Compared with positive drugs, the excretion of phenol red of NRG-NSps (30 mg/kg) increased 1.1-fold (1.45 vs. 1.17 μg/mL), and the expectorant effect was slightly increased but no significant difference was shown (*p* > 0.05).

## 3. Materials and Methods

### 3.1. Animals

Kunming mice (20 ± 2 g, 6 to 8 weeks old) were purchased from Vital River Laboratory Animal Technology Co., Ltd. (Beijing, China). The animals were raised under standard laboratory conditions and conducted according to the ethical and regulatory guidelines approved by the Animal Ethics Committee of Peking Union Medical College (Beijing, China), the ethical approval number of this study is SLXD-20191231005.

### 3.2. Materials

Naringenin (NRG, purity > 98%) was purchased from Aladdin Bio-Chem Technology Co., Ltd. (Shanghai, China). Tocopherol polyethylene glycol vitamin E succinate (TPGS; batch number: 20121203) was purchased from Xi′an Healthful Biotechnology Co., Ltd. (Xi′an, China). Dextromethorphan hydrobromide tablets were purchased from Guangzhou Baiyunshan Guanghua Pharmaceutical Co. Ltd. (Guangzhou, China). Ambroxol hydrochloride tablets were purchased from Shanghai Boehringer Ingelheim Pharmaceutical Co., Ltd. (Shanghai, China). All other reagents and solvents were purchased with analytical reagent grade.

### 3.3. Preparation of Naringenin Nanosuspension (NRG-NSps)

NRG-NSps were prepared using a miniaturized media-milling method [39,40]. The specific operation was undertaken as follows: TPGS (10 mg) was completely dissolved in 2 mL water under ultrasonic conditions. Next, NRG (40 mg) was ultrasonically dispersed in a vial containing 2 mL water. The TPGS aqueous solution was added to the vial, the vial was shaken, and then the drug was uniformly ultrasonically dispersed. After adding a stir bar and 10 g zirconia balls (0.4 to 0.6 mm in diameter), the mixed system was stirred (300 rpm min^−1^) at 50 °C. After 2 h stirring, the NRG-NSps solution was obtained.

### 3.4. Particle Size and Morphology

The mean hydrodynamic diameter, polydispersity index (PDI), and zeta potential of NRG-NSps were determined by dynamic light scattering (DLS, Zetasizer Nano ZS; Malvern Instruments, Malvern, UK) at 25 °C. Each sample was measured three times.

NRG-NSps (10 μL, 100 μg/mL) was placed on a 300-mesh copper sheet, and then negatively stained with 2% (*w*/*v*) uranyl acetate for 30 s [41]. After air-drying, the samples were measured by transmission electron microscope (JEOL Ltd., Tokyo, Japan) at 120 kV.

### 3.5. Drug-Loading Content (DLC) of NRG-NSps

The concentration of NRG was measured using high-performance liquid chromatography (HPLC; UltiMate 3000; DIONEX, Sunnyvale, CA, USA) at 25 °C [42]. A Waters Symmetry C18 column (250 mm × 4.60 mm, 5 μm) was utilized to separate the samples. The eluent was 70% chromatographic methanol containing 30% water (*v*/*v*) with a flow rate of 0.8 mL/min, the detection UV wavelength was designed as 388 nm, and the injection volume was 20 μL.

Lyophilized NRG-NSps powder was weighed (W) and dissolved in methanol (V). The concentration of NRG was determined by HPLC (C). The DLC was calculated using Equation (1):DLC (%) = V × C/W × 100%(1)

### 3.6. Stability of NRG-NSps

#### 3.6.1. Stability in Media

NRG-NSps was co-incubated with 0.9% NaCl, 5% glucose, PBS (pH 7.4), and artificial gastric and intestinal fluid at 37 °C separately. DLS was applied to determine the particle size and PDI value at 0, 2, 4, 6, and 8 h [43]. Each experiment was performed in triplicate.

#### 3.6.2. Storage Stability

The storage stability test was performed at 4 °C for 1 month [39]. The particle size and appearance of the formulation of NRG-NSps were determined on day 1, 7, 14, 21, and 30.

#### 3.6.3. Lyophilization Stability

Lyoprotectants were added to NRG-NSps solution at a concentration of 0.5% (*w*/*v*), and then lyophilized for 24 h after freezing at −80 °C (ALPHR 2–4 LD plus, CHRIST, Germany) [44]. The white powder was dispersed in 1 mL deionized water, and the particle size and PDI values were recorded.

### 3.7. NRG Release Behaviour

One mL NRG-NSps, NRG DMSO solution, and free NRG (dispersed in deionized water containing 0.5% CMC-Na) was added to a dialysis bag (MWCO: 8 000–14 000, Sigma-Aldrich, USA) separately and then placed in PBS solution at 37 °C [45]. The release medium (1.0 mL) was removed at 0.25, 0.5, 0.75, 1, 2, 4, 6, 8, 10, 12, 24, 36, 48, 72, 96, 120, 144, and 168 h. Meanwhile, 1.0 mL fresh media were added. After filtering through 0.22 μm membrane, the release media were analyzed by HPLC. The cumulative release (%) was calculated as the ratio of the weight of released NRG to total NRG. All experiments were performed in triplicate.

### 3.8. Cough Induction and Monitoring

Antitussive effects of NRG-NSps were estimated via mouse cough model [37]. Mice were placed in a 1 L glass chamber and exposed to 0.2 mL of 25% NH_4_OH for 3 min after acclimatization to laboratory conditions for three days. The mice were observed using video-observation equipment. The latent period and the cough frequency in 3 min were recorded, and mice presenting 8–20 times cough frequency were selected as test animals. These mice were randomly divided into 6 groups (10 mice per group) and orally administrated with saline (blank model control), dextromethorphan hydrobromide (15 mg/kg, positive control), NRG (30 mg/kg), and NRG-NSps (10, 30, and 50 mg/kg). After additional exposure to NH_4_OH, the number of coughing responses was measured over a period of 5 min.

### 3.9. Expectorant Properties of NRG-NSps

According to previous reports [46,47], kunming mice were divided into 6 groups after acclimatization to laboratory conditions for seven days (*n* = 8), including normal saline (blank model control), ambroxol hydrochloride (30 mg/kg, positive control), NRG (30 mg/kg), and NRG-NSps (10, 30, and 50 mg/kg, respectively). All groups were administrated daily for 7 d. All animals were treated with 5% phenol red in normal saline (1.25%, 10 mL/kg) via intraperitoneal injection on day 7, and after 30 min, mice were sacrificed by cervical dislocation, and their tracheae were removed and immediately placed into 2 mL normal saline. After adding 2 mL 5% sodium bicarbonate, the optical density was detected at 546 nm using a UV/visible spectrophotometer (UV2450; Shimadzu, Kyoto, Japan). To measure the excretion of phenol red in mouse trachea, linear regression analysis was performed in advance. The linearity parameter of the spectrophotometric was evaluated by measuring the relationship between the absorbance and the concentration of phenol red, where Y is the absorbance (AU) and X is the concentration (µg/mL) of phenol red. The regression line of the spectrophotometric response was obtained as the following equation (Equation (2)):Y = 0.1741x + 0.0168 (R^2^ = 0.999)(2)

## 4. Conclusions

To enhance the bioavailability and therapeutic efficacy of NRG, in this study, the miniaturization media-milling method was adopted to prepare NRG-NSps. After successful preparation, the particle size of NRG-NSps was approximately 216.9 nm, the PDI was 0.32, and the drug loading content was 66.7%, which presented flaky morphology. NRG-NSps showed good storage stability, media stability, and lyophilized resolution stability. In vivo experiments confirmed that 30 mg/kg oral administration of NRG-NSps could significantly reduce the frequency of acute cough in mice. Compared with the original drug NRG, the cough frequency of NRG-NSps was decreased by 31.8%, and the cough incubation period was enhanced by 42.5%. In addition, NRG-NSPs showed a good sputum-expelling effect as the positive drug, and the amount of phenol red secretion was enhanced by 42.1% and 23.9% compared with the original drug and positive drug, respectively. From these results, it was proven that NRG-NSps showed a good effect on relieving coughing and reducing phlegm. Furthermore, the pick ball grinding process is simple, easy to operate, and low cost, and its use at the industrial scale can be realized. Therefore, naringin nanosuspension has a potential clinical application.

## Figures and Tables

**Figure 1 molecules-27-00741-f001:**
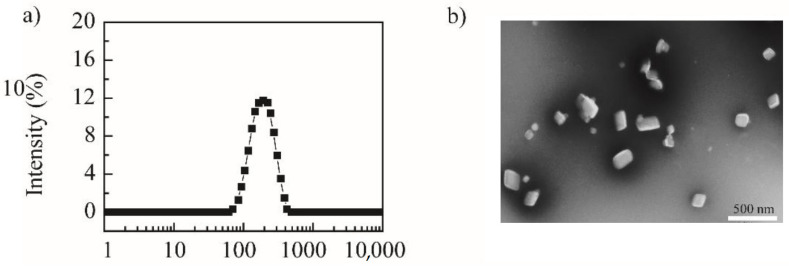
Particle size distribution curve (**a**) and TEM image of NRG-NSps (**b**) (scale bar: 500 nm).

**Figure 2 molecules-27-00741-f002:**
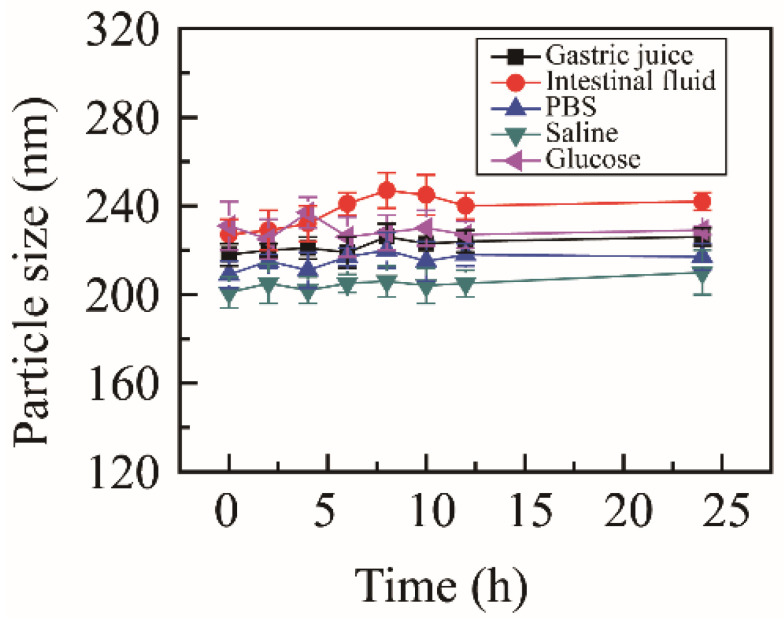
Particle size of NRG-NSps in different physiological media at 37 °C.

**Figure 3 molecules-27-00741-f003:**
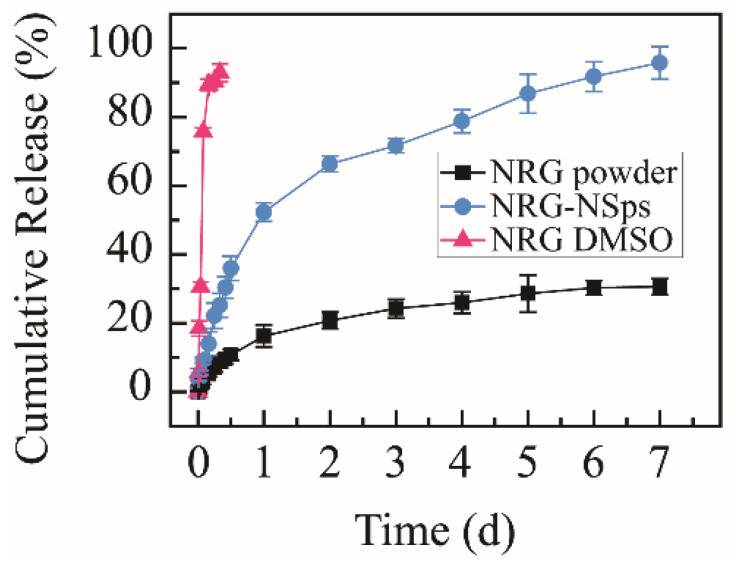
Cumulative release curves of NRG-NSps, NRG powder, and NRG DMSO solution in PBS (pH 7.4) at 37 °C.

**Figure 4 molecules-27-00741-f004:**
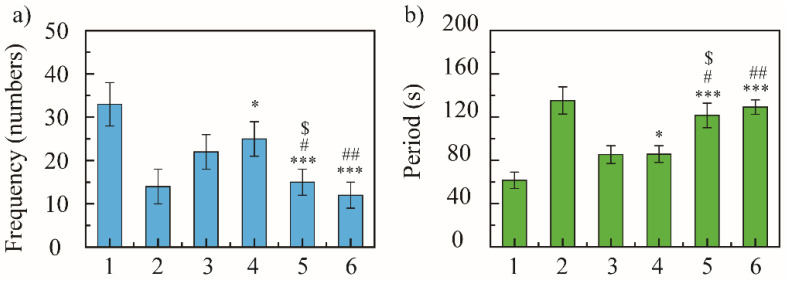
Cough-relieving effect of naringenin nanosuspension (NRG-NSps) on mice: cough frequency in 5 min (**a**) and cough incubation time (**b**), *n* = 10. 1: Saline group, 2: dextromethorphan hydrobromide group (15 mg/kg), 3: NRG group (30 mg/kg), 4: NRG-NSps group (10 mg/kg), 5: NRG-NSps group (30 mg/kg), and 6: NRG-NSps group (50 mg/kg). * *p* < 0.05 and *** *p* < 0.001 vs. saline group; ^#^
*p* < 0.05 and ^##^
*p* < 0.01 vs. NRG-NSps group (10 mg/kg); ^$^
*p* < 0.05 vs. NRG group (30 mg/kg).

**Figure 5 molecules-27-00741-f005:**
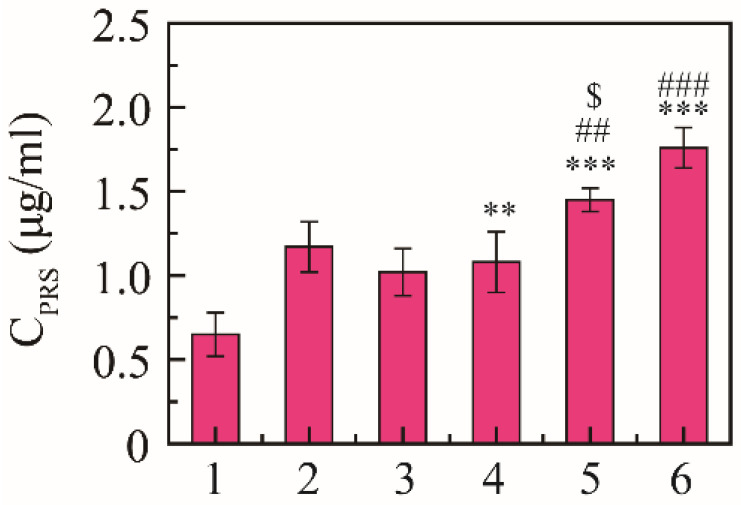
Expectorant activities of NRG-NSps (*n* = 8). 1: Saline group, 2: ambroxol hydrochloride group (15 mg/kg), 3: NRG group (30 mg/kg), 4: NRG-NSps group (10 mg/kg), 5: NRG-NSps group (30 mg/kg), and 6: NRG-NSps group (50 mg/kg). ** *p* < 0.01 and *** *p* < 0.001 vs. saline group; ^##^
*p* < 0.01 and ^###^
*p* < 0.001 vs. NRG-NSps group (10 mg/kg); ^$^
*p* < 0.05 vs. NRG group (30 mg/kg).

**Table 1 molecules-27-00741-t001:** Summary of particle size and zeta potential for NRG-NSps after storing at 4 °C for one month.

DLS Results	Time (d)			
	0	10	20	30
Size (nm)	216.9 ± 5.3	221.3 ± 6.2	223.3 ± 3.4	223.5 ± 3.5
PDI	0.32 ± 0.02	0.27 ± 0.01	0.26 ± 0.04	0.27 ± 0.03
Zeta potential (mV)	−1.08 ± 1.37	−1.41 ± 0.33	−0.16 ± 0.31	−0.28 ± 2.01

## Data Availability

Not applicable.

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
