# Peer review of "Preparation of Naringenin Nanosuspension and Its Antitussive and Expectorant Effects"

_molecules, 2022, doi:10.3390/molecules27030741_

Round 1
Reviewer 1 Report
Review comments to the author
Title: ''Preparation of naringenin nanosuspension and its antitussive 2 and expectorant effects''.
Manuscript ID: molecules-1509631.
Abstract:
1- Page 1, Line 15: The abbreviated term ''TPGs'' should be mention in full name for its first appearance followed by the abbreviated name.
- Introduction:
1- Page 2, Line 48: The two citations 12 and 13 should be written correctly as follows [12, 13].
2- Page 2, Line 65: The word ''in vivo'' should be written in italic font.
3- Page 2, Line 75: The two citations 23 and 24 should be written correctly as follows [23, 24].
4- Page 2, Line 82: Reference No. 28 appeared before reference 27; references in the text must be rearranged in numerical order.
- Results and Discussion
1- Page 3, Line 105: The sub-title ''2.2 Stability of NRG‑NSps'' should be typed as ''2.2. Stability of NRG‑NSps''.
2- Page 6, Line 207: The results in section ''2.7. Expectorant assay'' should be supported by suitable recent citations.
- Materials and Methods
1- The following sections should be supported by suitable recent citations:
- 3.3. Preparation of naringenin nanosuspension (NRG-NSps).
- 3.4. Particle size and morphology.
- 3.5. Entrapment efficiency (EE) and drug-loading content (DLC) of NRG-NSps.
- 3.6. Stability of NRG‑NSps.
- 3.7. NRG release behaviour.
- 3.8. Cough induction and monitoring.
Abbreviations:
- List of abbreviations should be inserted by the end of the manuscript before references.
References:
- In reference No. 8, add volume number.
- In reference No. 10, Journal name should be typed in full form.
- In reference No. 11, Journal name should be typed in proper way (don't use uppercase style).
- In reference No. 13, add page numbers.
- In reference No. 15, Journal name should be typed in full form.
- In reference No. 16, Journal name should be typed in full form.
- In references No. 21 and 22, complete page numbers.
- In reference No. 28, Journal name should be typed in full form.
- In reference No. 32, the words '' in vitro, in vivo '' should be written in italic fonts.
- In reference No. 34, Journal name should be typed in proper way as (Journal of Ethnopharmacology).
- In reference No. 35, Journal name should be typed in proper way as (Journal of Pharmacological and Toxicological Methods).
- In reference No. 37, Journal name should be typed in proper way as (Pakistan Journal of Pharmaceutical Sciences).
- In reference No. 39, add volume and page numbers.
- In reference No. 40, Journal name should be typed in proper way as (Journal of Ethnopharmacology).
- In reference No. 41, Journal name should be typed in proper way as (Experimental and Therapeutic Medicine).
- In reference No. 42, Journal name should be typed in proper way as (Environmental Toxicology and Pharmacology).
- In reference No. 43, Journal name should be typed in proper way as (Molecular and Cellular Biochemistry).
- In reference No. 44, Journal name should be typed in proper way as (Journal of Ginseng Research).
Author Response
Abstract:
1- Page 1, Line 15: The abbreviated term ''TPGs'' should be mention in full name for its first appearance followed by the abbreviated name.
The full name of TPGs is tocopherol polyethylene glycol succinate, which has been added in the revised manuscript.
Introduction:
1- Page 2, Line 48: The two citations 12 and 13 should be written correctly as follows [12, 13].
These citations have been corrected.
2- Page 2, Line 65: The word ''in vivo'' should be written in italic font.
The word “in vitro” and “in vivo” have been rewritten in italic font.
3- Page 2, Line 75: The two citations 23 and 24 should be written correctly as follows [23, 24].
These citations have been corrected.
4- Page 2, Line 82: Reference No. 28 appeared before reference 27; references in the text must be rearranged in numerical order.
These citations have been corrected.
- Results and Discussion
1- Page 3, Line 105: The sub-title ''2.2 Stability of NRG‑NSps'' should be typed as ''2.2. Stability of NRG‑NSps''.
The sub-title has been corrected.
2- Page 6, Line 207: The results in section ''2.7. Expectorant assay'' should be supported by suitable recent citations.
The relative reference has been added.
- Materials and Methods
1- The following sections should be supported by suitable recent citations:
- 3.3. Preparation of naringenin nanosuspension (NRG-NSps).
- 3.4. Particle size and morphology.
- 3.5. Entrapment efficiency (EE) and drug-loading content (DLC) of NRG-NSps.
- 3.6. Stability of NRG‑NSps.
- 3.7. NRG release behaviour.
- 3.8. Cough induction and monitoring.
The relative reference has been added.
Abbreviations:
- List of abbreviations should be inserted by the end of the manuscript before references.
NRG Naringenin
NRG-NSps Naringenin nanosuspensions
TPGS Tocopherol polyethylene glycol succinate
PVP Polyvinylpyrrolidone
FDA Food and Drug Administration
TEM Transmission electron microscopy
DLS Dynamic light scattering
PDI Polydispersity index
References:
- In reference No. 8, add volume number.
- In reference No. 10, Journal name should be typed in full form.
- In reference No. 11, Journal name should be typed in proper way (don't use uppercase style).
- In reference No. 13, add page numbers.
- In reference No. 15, Journal name should be typed in full form.
- In reference No. 16, Journal name should be typed in full form.
- In references No. 21 and 22, complete page numbers.
- In reference No. 28, Journal name should be typed in full form.
- In reference No. 32, the words '' in vitro, in vivo '' should be written in italic fonts.
- In reference No. 34, Journal name should be typed in proper way as (Journal of Ethnopharmacology).
- In reference No. 35, Journal name should be typed in proper way as (Journal of Pharmacological and Toxicological Methods).
- In reference No. 37, Journal name should be typed in proper way as (Pakistan Journal of Pharmaceutical Sciences).
- In reference No. 39, add volume and page numbers.
- In reference No. 40, Journal name should be typed in proper way as (Journal of Ethnopharmacology).
- In reference No. 41, Journal name should be typed in proper way as (Experimental and Therapeutic Medicine).
- In reference No. 42, Journal name should be typed in proper way as (Environmental Toxicology and Pharmacology).
- In reference No. 43, Journal name should be typed in proper way as (Molecular and Cellular Biochemistry).
- In reference No. 44, Journal name should be typed in proper way as (Journal of Ginseng Research).
All the references have been check carefully and listed with the Molecules style.
Reviewer 2 Report
The ms demonstrated the application of the method to prepare nanosuspension and the information generated is useful for the pharmaceutical area.
- There was only one latest ref in 2020. Please see if there is any update in the study area.
- Line 271-274 told the testing temp were 4 and 25C but only 4C data were shown in Table 1. Please clarify.
- Stability test of one month seem not sufficient for practical application. It i suggest to increase the length of stability testing.
Author Response
1. There was only one latest ref in 2020. Please see if there is any update in the study area.
Several latest ref. in 2021 have been added in the revised manuscript.
2. Line 271-274 told the testing temp were 4 and 25C but only 4C data were shown in Table 1. Please clarify.
We are sorry for this mistake. The storage stability was only evaluated at 4 °C. The description of 4 and 25 °C is a clerical error, and has been corrected in this revised manuscript.
3. Stability test of one month seem not sufficient for practical application. It i suggest to increase the length of stability testing.
Thanks very much for your comment. Until now, the NRG-NSps has been stored at 4 °C over 6 month, the appearance of NRG-NSps solution does not show significant difference, no turbidity or precipitation is observed, the particle diameter measured by dynamic light scattering is 241 nm. Besides, NRG-NSps solution can be lyophilized to powder for long time storage.